# COVID-19 and Mental Distress and Well-Being Among Older People: A Gender Analysis in the First and Last Year of the Pandemic and in the Post-Pandemic Period

**DOI:** 10.3390/geriatrics10010005

**Published:** 2025-01-03

**Authors:** M. Pilar Matud

**Affiliations:** Department of Clinical Psychology, Psychobiology, and Methodology, Universidad de La Laguna, 38200 San Cristobal de La Laguna, Spain; pmatud@ull.edu.es

**Keywords:** Coronavirus disease-2019 (COVID-19), older people, gender, mental distress, well-being, affect balance, life satisfaction, thriving, social support, stressful events

## Abstract

The Coronavirus disease 2019 (COVID-19) pandemic seriously threatened the health and well-being of the population. This study aims to investigate the relevance of the COVID-19 pandemic on the stress, mental distress, and well-being of older people in Spain. The design was quantitative repeated cross-sectional. The sample was non-probability and consisted of 1436 persons from the general population divided into two groups: (1) the study group, composed of 718 women (61.3%) and men aged 60 to 89; (2) the comparison group, composed of the same number of women and men aged 30 to 45. All were assessed in three phases of the COVID-19 pandemic: the first pandemic year, the last pandemic year, and the post-pandemic period. The results showed that during the first year of the pandemic, the prevalence of mental distress was higher in older women (50%) than in older men (37.2%), while the rates in the comparison group were 57.2% for women and 53.2% for men. In the post-pandemic period, the rates of mental distress were 30.2% for older women and 29.8% for older men while in the comparison group, the rates were 48.5% for women and 26.5% for men. No significant differences in well-being were found between the groups or between the different phases of the pandemic. The most common stressors reported by older people were illness and death of family and/or loved ones, followed by personal illness. In the post-pandemic period, more stressful events and lower stress resilience were found to predict mental distress in older women and men. Greater perceived vulnerability to infection was another important predictor for women. Low self-esteem and younger age were also predictors of mental distress for men. High self-esteem, high social support, greater stress resilience and fewer stressful events were predictors of well-being for both genders. The results of this study are relevant for the design of policies, programs, and strategies to improve the health and well-being of older people.

## 1. Introduction

Coronavirus disease 2019 (COVID-19), caused by SARS-CoV-2, emerged in late 2019 and spread rapidly worldwide. On 11 March 2020, it was declared a global pandemic by the World Health Organization (WHO) [1]. The WHO declared the end of COVID-19 as a public health emergency of international concern on 5 May 2023. However, it warned that this did not mean that COVID-19 was no longer a global health threat [2]. COVID-19 continues to be a global public health priority [2], and its past and continuing impact continues to be an important area of research [3]. After the end of the pandemic, SARS-CoV-2 has become endemic, with phases of flare-ups, and COVID-19 disease can still have a significant clinical impact [4]. Although many cases are asymptomatic [4], the symptoms of COVID-19 vary in severity from very mild to critical [5]. Age is a key determinant of severe disease and death from COVID-19 [6], and older patients with multimorbidity and frailty are particularly vulnerable [4]. There is evidence that older patients hospitalized for COVID-19 have a higher risk of mortality and severe COVID-19 outcomes. They are more likely to die within 30 days of hospitalization than non-COVID-19 patients [6]. Although further research is needed, it is considered that COVID-19 infection is likely to be a risk factor for the development of new-onset dementia in older adults over time [7]. And results from a recent systematic review of patients aged 65 years and older suggest an association between severe COVID-19 and cognitive impairment beyond the acute phase of illness [8].

There is evidence that COVID-19 has long-term physical and mental sequelae, although there is uncertainty about their prevalence, persistence, and predictors [9]. Symptoms, disability, and financial problems are prevalent in patients hospitalized with COVID-19 6 months after discharge [10]. A significant number of patients who recover from COVID-19 infection continue to have persistent symptoms or develop new symptoms that interfere with daily activities beyond the initial acute phase [11,12]. Fatigue and cognitive deficits are the most common sequelae after SARS-CoV-2 infection [13]. These problems improve over time, but about half of the patients experience symptoms that persist for up to two years [13]. According to the WHO, it is estimated that about 6% of symptomatic infections result in post-COVID-19 condition, which suggests that hundreds of millions of people require long-term management [14].

The COVID-19 pandemic resulted in a substantial increase in mortality in most countries of the world, although there are large differences in the estimates of excess deaths across the six WHO regions [15]. The pandemic has affected people from all continents, countries, races, and socioeconomic groups. It has become one of the greatest public health crises of a generation [16]. Nevertheless, the pandemic has not only affected health but also the environment, economy, education, and human psychology [17,18], with severe and widespread consequences for health systems and societies [19]. In the early stages of the COVID-19 pandemic, there was no effective treatment or vaccine available, and the mortality rate was high [20]. This, together with the rapid spread of the virus and the severity of its symptoms, led many governments to implement strict measures to control the transmission of the virus and protect the health and safety of their citizens. These measures included lockdowns, closing schools and workplaces, restricting travel, closing borders, and social distancing. Such measures have affected day to day life [21] and had important repercussions not only on the well-being of the population [22], as people had to limit their social and leisure activities and cope with the pressures of everyday life, but also on the economy and employment [23].

The older population was one of the most vulnerable groups during the COVID-19 pandemic [24,25,26], requiring significant efforts to prevent contamination, as the first fatal cases of the COVID-19 outbreak occurred mainly in the elderly [27]. There is evidence that the disease was more severe and even fatal in geriatric cases and those with chronic diseases [28]. Older adults were among the first groups to face restrictions on face-to-face contact [25]. Measures to contain the spread of the virus, such as social distancing and staying at home, had a major impact on older people. They disrupted their daily routines, reduced their physical activity, their contact with friends and family, and their participation in cultural events and other leisure activities [29,30,31,32]. Although reducing viral transmission is critical, physical distance is associated with negative psychosocial outcomes, such as higher rates of anxiety and depression [25,33], and changes in daily routines have implications for health and well-being [30]. Social isolation has been negatively associated with subjective well-being [34] and increased feelings of loneliness [33,34], and loneliness has also been associated with poor well-being [35]. Although many older people reported feeling lonely during the pandemic [36,37], loneliness was associated with sociodemographic factors, including being female [35,36,37] and living alone [36].

There is evidence that the COVID-19 pandemic had a negative impact on the mental health of older people compared with pre-pandemic levels, with increases in stress and mental distress [38,39,40]. However, this increase was not uniform and varied by demographic and personal and social characteristics. Women [41,42,43], persons under 60 years of age, uncertainty whether infected with SARS-CoV2, and a higher perceived risk of infection were associated with higher levels of mental distress [41]. Being married has been found to be a protective factor for well-being in older people [44]. And while economic concerns and financial strain predicted greater distress, education and income had little predictive power, according to a systematic review of the relationships between socioeconomic conditions and emotional problems in 98 countries during COVID-19 [45]. Evidence indicates that psychological responses to potentially traumatic events associated with the COVID-19 pandemic were heterogeneous [46,47]. Trajectories of mental distress varied significantly by resilience [46]. Other factors that have been shown to protect mental health and well-being during the COVID-19 pandemic include social support [48,49,50,51] and self-esteem [49,50,51,52].

While there have been many studies of mental health during the COVID-19 pandemic, most have focused on the early days of the pandemic, when there were numerous restrictions to control the spread of the virus, the population was not vaccinated, and coronavirus disease was more severe. Few studies of mental health include data from 2022 onwards [53]. In addition, although there is evidence that the social, economic, and health crisis caused by the COVID-19 pandemic affected men and women differently [54,55], many previous studies were not conducted from a gender perspective and did not present all data disaggregated by gender.

In more than three years between the WHO’s declaration of a pandemic on 11 March 2020 [1] and its declaration of the end of COVID-19 as a global health emergency on 5 May 2023 [2], much has changed for people, both in terms of the health risks and restrictions imposed to contain the spread of the virus, and in terms of the social and economic consequences. Many efforts have been made to significantly reduce the impact of COVID-19 on morbidity and mortality, including the development of effective COVID-19 vaccines, treatments, and diagnostic tests [14]. By the end of 2021, 58% of the global population had received some vaccine and 73–81% of the world’s population were likely to have been infected, vaccinated, or both [56]. The death rate was reduced due to the administration of more than 12.8 billion doses of vaccines globally and the decreased pathogenicity of SARS-CoV-2 variants [57]. For this reason, the data collection in the present work was carried out in three different time periods (which we will call “phases”): (1) The first year of the pandemic, between June and December 2020, a phase in which the disease was severe, mortality due to SARS-CoV-2 was high, and there were numerous restrictions to limit the spread of the virus. (2) During the last pandemic year, between February 2022 and February 2023, when most of the Spanish population was vaccinated, COVID-19 disease was not severe in most people, and there were virtually no social restrictions due to the COVID-19 pandemic. (3) In the post-pandemic period, between October 2023 and May 2024, a period of SARS-CoV-2 endemicity with phases of flare-ups [4]. Since there is no widely accepted quantitative definition of the end of a pandemic such as COVID-19 [56], in this study, the post-pandemic period is considered to be the period following the declaration of the end of COVID-19 as a public health emergency of international concern by the WHO on 5 May 2023 [2] and the declaration of the end of the health crisis situation caused by COVID-19 in Spain by the Council of Ministers in its meeting on 4 July 2023, upon the proposal of the Minister of Health [58]. The data collection started three months after the latter date, to give the population time to become aware of the new situation.

The general objective of this study is to investigate the relevance of the COVID-19 pandemic on the mental distress and well-being of older people in Spain. The first aim is to know the presence of stress and mental distress during the three phases of the pandemic studied and to know if there are differences in mental distress, well-being, self-esteem, social support, and stressful events in older people according to gender and the phase of the pandemic. These analyses will also be conducted on a comparison group of adults (aged 30–45) to identify differences and/or similarities between older and younger adults. The second aim of this study is to determine the relevance of sociodemographic variables, perceived vulnerability to infection, and the number of stressful events as risk factors and stress resilience, self-esteem, and social support as protective factors against mental distress and promoting well-being in older women and men in the post-pandemic period.

## 2. Materials and Methods

### 2.1. Study Design and Participants

The current study used a quantitative repeated cross-sectional design. Data were collected through an online survey using a Google Form. All participants were volunteers, recruited through the researchers’ social network and through undergraduate and postgraduate students enrolled in various Spanish universities, who received course credit for their participation in the data collection. The sample is non-probabilistic and consists of 1436 individuals divided into two groups: (1) study group (“older group”), composed of 718 women (61.3%) and men aged between 60 and 89 years; (2) comparison group (“adult group”), consisting of the same number of women and men as the older group aged between 30 and 45 years. All individuals lived in Spain and were assessed in three phases related to the COVID-19 pandemic: (1) during the first year of the pandemic (*n* = 702), between June and December 2020; (2) during the last pandemic year (*n* = 326), between February 2022 and February 2023; (3) in the post-pandemic period (*n* = 408), between October 2023 and May 2024. The individuals who responded at each phase are different, as this is a cross-sectional design. The comparison group was selected by controlling for the following: (1) that they were aged between 30 and 45; (2) that no statistically significant differences existed in the level of education compared to the older group; (3) that the number of women and men in each of the three phases of the study was equal to the number of older women and men. Table 1 shows the main characteristics of the older people. As can be seen, although there was diversity in their sociodemographic characteristics, they were more likely to have a university education, to be retired, and to be married or living with a partner. Women and men did not differ in age or occupation, but they did differ in the other sociodemographic variables. For men, the number of children ranged from 0, which occurred in 6.5% of cases, to 10, which occurred in only one man. For women, the range was from 0 children, which occurred in 19.7% of women, to 8 children, which occurred in one case. As can be seen in Table 1, the mean number of children was slightly higher for men than for women.

This study is part of a larger investigation into the impact of the COVID-19 pandemic in Spain. Being 60 years of age or older was the condition for inclusion in the study group. Once this group was formed, we proceeded to select the comparison group by randomly selecting participants from a much larger database of 8578 individuals (65.2% women) between the ages 19 and 58 with different sociodemographic characteristics, with the following requirements: they should be between 30 and 45 years old and have a similar level of education to the study group. The comparison group also had the same number of women and men at each of the three phases as the study group.

### 2.2. Variables and Instruments

**The 12-item General Health Questionnaire (GHQ-12)** [59]. This is a 12-item screening instrument and is a commonly used measure of mental distress [60]. An example item is “felt constantly under strain”. Scores were coded using the Likert scale (0-1-2-3) and the standard GHQ scoring system (0-0-1-1). The Likert score was used in all analyses, except for the discrimination of distressed from non-distressed cases, where the standard GHQ score was used. The GHQ threshold for discriminating distressed from non-distressed cases was ≥ 4, as this is the most appropriate threshold according to Lundin et al. [61], with sensitivity = 81.7 and specificity = 85.4. For the current sample, the Cronbach’s α of the 12 items scored according to the Likert method was 0.88, while it was also 0.89 according to the standard GHQ method.

**Stressful events** [51,62,63]. Participants are asked whether they have experienced the following events and/or losses: (1) job loss; (2) financial difficulties; (3) serious arguments with their partner; (4) serious family discord; (5) illness of family members or other loved ones; (6) death of one or more family members and/or other loved ones; (7) personal illness; (8) other event(s) and/or loss(es) experienced. The presence of each event was scored as one, and the total number of events was calculated as the sum of the number of events for which the answer was yes.

**Scale of Positive and Negative Experiences (SPANE)** [64]. This is a scale consisting of twelve items, six of which assess positive feelings (SPANE-P) and six of which assess negative feelings (SPANE-N). An example item for positive feelings is “pleasant” and for negative feelings, “angry”. Each item is scored on a scale ranging from 1 (very rarely or never) to 5 (very often or always). The positive and negative scales are scored independently, and the two scores are combined to measure affect balance (SPANE-B). Affect balance is a construct depicting the balance of positive to negative emotions. Individuals with higher scores have positive affect that outweighs the experience of negative affect [65]. For the current sample, Cronbach’s alpha was 0.92 for positive feelings, 0.86 for negative feelings, and 0.90 for affect balance.

**The Brief Inventory of Thriving (BIT)** [66]. The BIT is a 10-item inventory designed to assess psychological well-being. An example item is “My life has a clear sense of purpose”. Items are scored on a scale from 1 (strongly disagree) to 5 (strongly agree). For the present sample, Cronbach’s alpha was 0.89.

**Satisfaction with Life Scale (SWLS)** [67]. This is a 5-item scale that assesses overall satisfaction with life, which is considered to be the cognitive component of subjective well-being. An example item is “I am satisfied with my life”. Each item is rated on a scale from 1 (strongly disagree) to 7 (strongly agree). The SWLS is a widely used instrument that has been validated in many countries and with samples of different ages, including older people [68]. Cronbach’s alpha for the present sample was 0.87.

**Social Support Scale (SSS)** [69]. This scale consists of 12 items that assess perceived social support in both instrumental and emotional domains. An example item is “Someone who comforts you when you are upset”. Items are scored on a 4-point scale ranging from 0 (never) to 3 (always). Cronbach’s alpha for the 12 items comprising the scale was 0.94 for the current sample.

**The Rosenberg Self-Esteem Scale (RSES)** [70]. The RSES is a 10-item scale that assesses global self-esteem. An example item is “I take a positive attitude toward myself”. All items are scored on a 4-point scale ranging from 0 (strongly agree) to 3 (strongly disagree). In the sample of the present study, Cronbach’s alpha was 0.84.

**The Brief Resilience Scale** [71]. This is a scale consisting of 6 items. It assesses the person’s ability to bounce back or recover from stress. An example item is “It does not take me long to recover from a stressful event”. The response scale is a five-point scale. It ranges from 1 (strongly disagree) to 5 (strongly agree). For the current sample, Cronbach’s alpha coefficient was 0.70.

**Perceived Vulnerability to Disease Questionnaire (PVDQ)** [72,73]. The PVDQ consists of 15 items measuring beliefs related to personal susceptibility to infectious diseases. It also measures perceptions and behaviors in situations with a potential risk of pathogen transmission. An example item is “If an illness is ’going around’, I will get it”. The response scale is a seven-point Likert scale from 1 (strongly disagree) to 7 (strongly agree). For the current sample, Cronbach’s alpha coefficient was 0.76.

The following **sociodemographic variables** were assessed: gender (women, men, other), age, marital status, number of children, education, and occupation. The level of education had five possible responses. These ranged from 0 (no formal education) to 4 (university education). As very few people responded to the gender category ’other’, they were not included in this study.

### 2.3. Data Analysis

Descriptive analyses were used to examine the demographic characteristics of the participants. To analyze differences between groups, the t-test was used for quantitative variables and the chi-square test for categorical variables. Internal consistency was calculated using Cronbach’s alpha coefficient. Normality for continuous variables was analyzed using graphical and numerical methods by analyzing kurtosis and skewness [74]. The homogeneity of variance was checked using Levene’s test. To address the first aim of the work, which focused on whether there were differences in mental distress, well-being, self-esteem, social support, and stressful events in older people by gender and by phase of the pandemic, and whether such differences were also found in adult persons, between-subjects analyses of variance (ANOVA) were conducted. In the first set of ANOVAs, the factor variables were gender (men, women) and pandemic phase (first year of the pandemic, last pandemic year, post-pandemic period). These analyses were conducted independently for the older group and for the adult group. In the second group of ANOVA analyses, the factor variables were pandemic phase (first year of the pandemic, last pandemic year, post-pandemic period) and group (older, adult). These analyses were performed independently for women and men. Dependent variables in all ANOVAs were mental distress, negative feelings, positive feelings, affect balance, life satisfaction, thriving, self-esteem, social support, and the number of stressful events.

To address the second study aim, which was to determine the relevance of sociodemographic variables, perceived vulnerability to infection, and the number of stressful events as risk factors, and stress resilience, self-esteem, and social support as protective factors against mental distress and promoting well-being in older women and men in the post-pandemic period, hierarchical multiple regression analyses were performed. In the first step (Model 1), sociodemographic variables (age, education, number of children, and whether they had a partner or not) were included. Age and the number of children were entered as quantitative variables. Education was included as an ordinal variable with 4 levels, ranging from 0 (no formal education) to 4 (university education). The presence or absence of a partner was a dummy variable with two levels: without a partner, the score was 0; with a partner, the score was 1. In step 2 (Model 2), perceived vulnerability to infection was added. In step 3 (Model 3), the number of stressful events was added. In step 4 (Model 4), stress resilience was added. In step 5 (Model 5), self-esteem and social support scores were added. The criterion to be predicted was mental distress in the first regression analyses and well-being in the second analyses. The well-being score was the sum of the affective balance, thriving, and life satisfaction scores. All statistical analyses were performed using IBM SPSS Statistics 29.0.

## 3. Results

### 3.1. Differences in Mental Distress, Well-Being, Self-Esteem, Social Support, and Number of Stressful Events by Gender, Phase of the Pandemic, and Age Group

The prevalence of mental distress among older people during the first year of the pandemic was 50% for women and 37.2% for men. These differences were statistically significant, χ^2^ (1, *N* = 351) = 5.08, *p* = 0.024). The prevalence among adults at this phase was 57.2% for women and 53.2% for men. These differences were not statistically significant. An analysis of differences in prevalence by age group showed statistically significant differences between older and adult men, χ^2^ (1, *N* = 224) = 5.28, *p* = 0.016), but not between older and adult women. During the last pandemic year, 36.7% of older women and 23.3% of older men experienced mental distress. Among adults, the prevalence during this phase was 43.0% for women and 30.5% for men. There were no statistically significant differences in the percentages by gender or age group. In the post-pandemic period, the prevalence of mental distress among older people was 30.2% for women and 29.8% for men. The prevalence for adults in this period was 48.5% for women and 26.5% for men, differences that were statistically significant, χ^2^ (1, *N* = 201) = 10.41, *p* = 0.001). During this period, there were no statistically significant differences in the prevalence of mental distress between older and adult men. However, there were statistically significant differences in the prevalence of mental distress between older and adult women, χ^2^ (1, *N* = 195) = 6.81, *p* = 0.009. In analyzing whether there were statistically significant differences in the prevalence of mental distress in the three phases of the pandemic studied, it was found that the differences between phases were statistically significant for older women, χ^2^ (2, *N* = 432) = 12.74, *p* = 0.002; for adult women, χ^2^ (2, *N* = 435) = 6.29, *p* = 0.04; and for adult men, χ^2^ (2, *N* = 272) = 17.93, *p* < 0.001; but not for older men.

Table 2 shows the means, standard deviations, and main results of the ANOVAs for the elderly (between 60 and 89 years) and Table 3 for the adults (between 30 and 45 years). The gender × phase interaction was not statistically significant in either group, as can be seen in both tables.

Among older participants (60–89 years), the main effect of gender was statistically significant when the dependent variable was mental distress (*p* < 0.001), negative feelings (*p* < 0.001), and affect balance (*p* = 0.03). When the dependent variable was mental distress, post hoc analyses with Games-Howell adjustment to determine which groups had statistically significant differences showed that such differences existed between women during the first year of the pandemic and the three groups of men. Women during the first year of the pandemic had greater mental distress than men during the first year of the pandemic (*p* = 0.009), than men during the last year of the pandemic (*p* = 0.002), and than men during the post-pandemic period (*p* = 0.004). In addition, women had more mental distress during the first year of the pandemic than during the post-pandemic period (*p* = 0.05). When the dependent variable was negative feelings, post hoc analyses with Scheffe adjustment revealed statistically significant differences between women and men only during the first year of the pandemic, with women having more negative feelings than men (*p* = 0.037). Post hoc analyses revealed no statistically significant differences between any of the groups when the dependent variable was affect balance.

The main effect of phase was statistically significant (*p* < 0.001) only when the dependent variable was the number of stressful events. Post hoc analyses with Scheffe adjustment showed that there were statistically significant differences between women in the last year of the pandemic and women in the post-pandemic period compared with men and women in the first year of the pandemic. Men in the first year of the pandemic reported fewer stressful events than women in the last year of the pandemic (*p* = 0.024) and in the post-pandemic period (*p* = 0.014). Women in the first year of the pandemic reported fewer stressful events than women in the last year of the pandemic (*p* = 0.001) and than women in the post-pandemic period (*p* < 0.001).

Among adults (30–45 years), the main effects of gender (*p* = 0.003) and phase (*p* = 0.003) were statistically significant when the dependent variable was mental distress. Post hoc analyses with the Games-Howell adjustment showed statistically significant differences between men in the post-pandemic period compared with men in the first pandemic year (*p* = 0.022), women in the first pandemic year (*p* < 0.001), and women in the post-pandemic period (*p* = 0.018); and between men in the last pandemic year and women in the first year of the pandemic (*p* = 0.021). Men in the post-pandemic period had less mental distress than the other groups, and men in the last pandemic year had less mental distress than women in the first pandemic year, as shown in Table 3. Only the main effect of gender was statistically significant (*p* < 0.001) when the dependent variable was negative feelings. Post hoc analysis with Scheffe adjustment showed that there were statistically significant differences between men in the post-pandemic period compared to women in the first year of the pandemic (*p* = 0.005) and in the post-pandemic period (*p* = 0.005). Men in the post-pandemic period had fewer negative feelings than women in the post-pandemic period and in the first year of the pandemic. When the dependent variable was positive feelings, the main effect of phase was statistically significant (*p* = 0.001). However, post hoc analyses with Scheffe adjustment revealed no statistically significant differences between any of the groups. The main effects of gender (*p* = 0.019) and phase (*p* = 0.004) were statistically significant when the dependent variable was affect balance. Post hoc analyses revealed statistically significant differences only between men during the post-pandemic period and women during the first year of the pandemic (*p* = 0.007). When the dependent variable was life satisfaction, the main effect of gender was statistically significant (*p* = 0.038). However, post hoc analyses with Scheffe adjustment revealed no statistically significant differences between the groups. When the dependent variable was thriving, the main effect of phase was statistically significant (*p* = 0.033). However, post hoc analyses with Scheffe adjustment revealed no statistically significant differences between the groups. When the dependent variable was the number of stressful events, post hoc analyses with Scheffe adjustment revealed statistically significant differences (*p* = 0.047) between women in the post-pandemic period and women in the first year of the pandemic, with women reporting a greater number of stressful events in the post-pandemic period.

To determine whether there were differences in the study variables between the older and adult groups in each of the phases analyzed, ANOVAs were also performed. In these analyses, the factors were age group (older, adult) and phase (first year of the pandemic, last pandemic year, post-pandemic period). Women and men were analyzed separately. Table 2 and Table 3 show the means and standard deviations for each group. For those variables where a statistically significant effect of age group was found in women, Figure 1 shows the changes. As the results of the post hoc comparisons according to the three phases studied have already been presented, these results will not be presented again.

In the women’s group, when the dependent variable was mental distress, the age group × phase interaction was not statistically significant, *F*(2, 874) = 1.49, *p* = 0.23. The main effects of age, *F*(2, 874) = 7.34, *p* = 0.007, η_p_^2^ = 0.008, and phase, *F*(2, 874) = 3.86, *p* = 0.021, η_p_^2^ = 0.009, were statistically significant. Post hoc analyses with Games-Howell adjustment showed statistically significant differences (*p* < 0.001) between older women in the post-pandemic period compared with adult women during the first year of the pandemic. Older women had less mental distress, as shown in Figure 1a.

When the dependent variable was negative affect, the age group × phase interaction was not statistically significant, *F*(2, 874) = 2.27, *p* = 0.10. The main effect of phase was also not statistically significant, *F*(2, 874) = 1.94, *p* = 0.14. The main effect of age was statistically significant, *F*(2, 874) = 22.38, *p* < 0.001, η_p_^2^ = 0.025. Post hoc analyses with Scheffe adjustment showed statistically significant differences between adult women in the first year of the pandemic compared with older women in the first year (*p* = 0.033), in the last year of the pandemic (*p* = 0.047), and in the post-pandemic period (*p* = 0.007). There were also statistically significant differences between adult women in the post-pandemic period compared with older women in the first year (*p* = 0.036), in the last year of the pandemic (*p* = 0.034), and in the post-pandemic period (*p* = 0.007). As can be seen in Figure 1b, older women had fewer negative feelings in all three phases of the pandemic than adult women in the first year of the pandemic and in the post-pandemic period.

When the dependent variable was positive feelings, the age group × phase interaction was not statistically significant, *F*(2, 874) = 2.50, *p* = 0.08. The main effect of age was also not statistically significant, *F*(2, 874) = 2.47, *p* = 0.12. The main effect of phase was statistically significant, *F*(2, 874) = 4.05, *p* = 0.018, η_p_^2^ = 0.009. When the dependent variable was affect balance, the age group × phase interaction was not statistically significant, *F*(2, 874) = 2.52, *p* = 0.08. The main effect of age was also not statistically significant, *F*(2, 874) = 3.25, *p* = 0.07. The main effect of phase was statistically significant, *F*(2, 874) = 3.32, *p* = 0.037, η_p_^2^ = 0.008. When the dependent variable was life satisfaction, the age group × phase interaction was not statistically significant, *F*(2, 874) = 1.06, *p* = 0.35. The main effect of age group was also not statistically significant, *F*(2, 874) = 1.34, *p* = 0.25, nor was the main effect of phase, *F*(2, 874) = 0.48, *p* = 0.62. When the dependent variable was thriving, the age group × phase interaction was not statistically significant, *F*(2, 874) = 0.69, *p* = 0.50. Neither the main effect of age group, *F*(2, 874) = 2.01, *p* = 0.16, nor the main effect of phase, *F*(2, 874) = 1.88, *p* = 0.15, were statistically significant. When self-esteem was the dependent variable, the age × phase interaction was not statistically significant, *F*(2, 866) = 0.76, *p* = 0.47. The main effect of age group was also not statistically significant, *F*(2, 866) = 0.65, *p* = 0.42, nor was the main effect of phase, *F*(2, 866) = 0.31, *p* = 0.73.

When the dependent variable was social support, there was no statistical significance for the age group x phase interaction, *F*(2, 874) = 1.79, *p* = 0.17. There was also no statistical significance for the main effect of phase, *F*(2, 874) = 0.47, *p* = 0.99. The main effect of age was statistically significant, *F*(2, 874) = 20.99, *p* < 0.001, η_p_^2^ = 0.023. Post hoc analyses with Scheffe adjustment showed the existence of statistically significant differences only between older women and adult women in the last year of the pandemic (*p* = 0.026). Older women had less perceived social support than adult women, as shown in Figure 1c. When the dependent variable was the number of stressful events, the age group × phase interaction was not statistically significant, *F*(2, 825) = 1.44, *p* = 0.24. The main effect of age was also not statistically significant, *F*(2, 825) = 0.97, *p* = 0.33. The main effect of phase was statistically significant, *F*(2, 825) = 20.66, *p* < 0.001, η_p_^2^ = 0.048. Post hoc analyses with Scheffe adjustment showed the existence of statistically significant differences (*p* < 0.05) between older women during the first year of the pandemic and all other groups except adult women during the first year of the pandemic. As shown in Figure 1d, older women reported fewer stressful events during the first year of the pandemic.

Figure 2 shows the changes as a function of age group among men for those study variables for which a statistically significant effect of age group was found. Means and standard deviations are presented in Table 1 and Table 2. Because the post hoc comparisons according to the three study phases have already been presented, these results will not be presented again.

In the men’s group, when the dependent variable was mental distress, the age group × phase interaction was not statistically significant, *F*(2, 550) = 2.99, *p* = 0.051. The main effects of age, *F*(2, 550) = 6.14, *p* = 0.013, η_p_^2^ = 0.011, and phase, *F*(2, 550) = 3.75, *p* = 0.024, η_p_^2^ = 0.013, were statistically significant. Post hoc analyses with Games-Howell adjustment showed statistically significant differences (*p* < 0.001) between adult men in the first pandemic year compared with older men in the first pandemic year (*p* = 0.018), in the last pandemic year (*p* = 0.005), and in the post-pandemic period (*p* = 0.011). Older men had less mental distress than adult men, as shown in Figure 2a.

When the dependent variable was negative feelings, the age group x phase interaction was not statistically significant, *F*(2, 550) = 2.85, *p* = 0.059. The main effect of phase was also not statistically significant, *F*(2, 550) = 0.71, *p* = 0.49. The main effect of age was statistically significant, *F*(2, 550) = 10.92, *p* = 0.001, η_p_^2^ = 0.019. Post hoc analyses with Scheffe adjustment showed statistically significant differences between adult men in the first year of the pandemic compared to older men in the first year (*p* = 0.013) and in the last year of the pandemic (*p* = 0.043). Older men had fewer negative feelings, as shown in Figure 2b.

When the dependent variable was positive feelings, the age group × phase interaction was not statistically significant, *F*(2, 550) = 0.64, *p* = 0.53. The main effect of age was also not statistically significant, *F*(2, 550) = 1.48, *p* = 0.22. The main effect of phase was statistically significant, *F*(2, 550) = 3.87, *p* = 0.022, η_p_^2^ = 0.014. When the dependent variable was affect balance, the age group × phase interaction was not statistically significant, *F*(2, 550) = 1.90, *p* = 0.15. Neither the main effect of age group, *F*(2, 550) = 1.40, *p* = 0.24, nor the main effect of phase, *F*(2, 550) = 2.25, *p* = 0.11, were statistically significant. When the dependent variable was life satisfaction, the age group × phase interaction was not statistically significant, *F*(2, 550) = 0.80, *p* = 0.45. Neither the main effect of age group, *F*(2, 550) = 0.41, *p* = 0.52, nor the main effect of phase, *F*(2, 550) = 0.80, *p* = 0.45, were statistically significant When the dependent variable was thriving, the age group × phase interaction was not statistically significant, *F*(2, 550) = 2.06, *p* = 0.13. The main effect of age group was also not statistically significant, *F*(2, 550) = 2.56, *p* = 0.11, nor was the main effect of phase, *F*(2, 550) = 0.73, *p* = 0.48. When the dependent variable was self-esteem, the age group × phase interaction was not statistically significant, *F*(2, 548) = 0.25, *p* = 0.78. The main effect of age group was not statistically significant, *F*(2, 548) = 0.31, *p* = 0.58, nor was the main effect of phase, *F*(2, 548) = 0.90, *p* = 0.41. When the dependent variable was social support, the age group × phase interaction was not statistically significant, *F*(2, 550) = 1.95, *p* = 0.14. The main effect of phase was also not statistically significant, *F*(2, 550) = 0.21, *p* = 0.81. The main effect of age was statistically significant, *F*(2, 550) = 7.13, *p* = 0.008, η_p_^2^ = 0.013. However, post hoc analysis with Scheffe adjustment revealed no statistically significant differences between any of the groups. When the dependent variable was the number of stressful events, the age group × phase interaction was not statistically significant, *F*(2, 538) = 1.76, *p* = 0.17. Neither the main effects of age group, *F*(2, 538) = 1.42, *p* = 0.23, nor phase, *F*(2, 538) = 2.57, *p* = 0.08, were statistically significant.

An analysis of the most common stressful events reported since the beginning of the COVID-19 pandemic showed that the most common event was the illness of family members and/or other loved ones, reported by 46.1% of the older people and 43.8% of the adults, differences that were not statistically significant. The second most common event reported by older people was the death of one or more family members and/or other loved ones. This occurred in 36% of older people and 26.8% of adults, differences that were statistically significant, χ^2^ (1, *N* = 1425) = 14.05, *p* < 0.001. Personal illness was next, reported by 28.8% of older people and 15.2% of adults, differences that were statistically significant, χ^2^ (1, *N* = 1410) = 37.76, *p* < 0.001. Financial difficulties were reported by 13.4% of the older people and 30.1% of the adults. These differences were statistically significant, χ^2^ (1, *N* = 1423) = 58.33, *p* < 0.001. Serious family discord was reported by 12.8% of the older people and 20% of the adults. These differences were statistically significant, χ^2^ (1, *N* = 1423) = 13.43, *p* < 0.001. Serious arguments with their partner were reported by 8.9% of older people who had a partner and 26.1% of adults who had a partner. These differences were statistically significant, χ^2^ (1, *N* = 918) = 46.25, *p* < 0.001. The least common event among older people was job loss, which occurred in 5.1% of older people and 15.4% of adults. These differences were statistically significant, χ^2^ (1, *N* = 1421) = 40.99, *p* < 0.001. In total, 16.4% of the older people reported other stressful events, while the proportion of adults reporting other events was 6.3%. The differences were statistically significant, χ^2^ (1, *N* = 1436) = 36.88, *p* < 0.001.

When analyzing whether there were differences in the frequency of each event among older people according to gender and the phase of the pandemic studied, it was found that there were no statistically significant differences for job loss, financial difficulties, serious arguments with partner, and serious family discord. There were statistically significant differences for the illness of family members and/or other loved ones, χ^2^ (5, *N* = 711) = 81.58, *p* < 0.001. This event was less common in the first year of the pandemic, when it was reported by 30.4% of men and 28.5% of women, than in later phases, when it was reported by 60.7% of men and 65% of women in the last year of the pandemic and by 56.7% of men and 66.7% of women in the post-pandemic period. There were also statistically significant differences for the death of one or more family members and/or other loved ones, χ^2^ (5, *N* = 713) = 45.76, *p* < 0.001. This event was reported less frequently in the first year of the pandemic, when it was reported by 28.3% of men and 21.7% of older women, than in later phases. It was reported by 44.3% of men and 47.5% of women in the last year of the pandemic, and by 49.5% of men and 48% of women in the post-pandemic period. There were also statistically significant differences for personal illness, χ^2^ (5, *N* = 706) = 21.38, *p* = 0.001. Such an event was reported by 22.5% of men and 20.9% of older women in the first year of the pandemic, by 33.3% of men and 36.6% of women in the last year of the pandemic, and by 29.8% of men and 42% of women in the post-pandemic period. There were also statistically significant differences for other events, χ^2^ (5, *N* = 718) = 12.51, *p* = 0.028. Other events were reported by 18.6% of older men and 20.6% of older women in the first year of the pandemic, by 14.8% of men and 14.7% of women in the last year of the pandemic, and by 5.8% of men and 18% of women in the post-pandemic period.

### 3.2. Predictors of Mental Distress and Well-Being in Older Women and Men in the Post-Pandemic Period

Table 4 presents the main results of the hierarchical regression analysis predicting mental distress in older women and in Table 5, in older men. In the women’s group, no sociodemographic variable predicted mental distress, whereas in men, age was a statistically significant predictor, with greater distress at younger ages. Perceived vulnerability to infection, added in Model 2, statistically significantly increased the prediction of mental distress, although its predictive value was much higher in women (R^2^ change = 0.24) than in men (R^2^ change = 0.05). In fact, this variable was no longer statistically significant in the men’s group when the number of stressful events (Model 3) was included in the regression equation. The number of stressful events significantly increased the prediction of mental distress in both groups. There was a statistically significant increase in the prediction of mental distress in both groups when stress resilience was added (Model 4). The addition of self-esteem and social support in Model 5 also statistically significantly increased the prediction of mental distress. However, the increase in predictive power was greater in the men’s group (R^2^ Change = 0.11) than in the women’s group (R^2^ Change = 0.05). In the final model, with all variables in the regression equation, it was found that for women, the main predictor of mental distress was the number of stressful events. This was followed by perceived vulnerability to infection and stress resilience. Older women who had experienced a higher number of stressful events, higher perceived vulnerability to infection, and lower stress resilience had more mental distress in the post-pandemic period. Among men, four predictors were statistically significant in the final model: the number of stressful events, self-esteem, age, and stress resilience. Older men with more mental distress in the post-pandemic period were those who had experienced a greater number of stressful events, those with lower self-esteem, those who were younger, and those with lower stress resilience. The percentage of total variance explained was 46% for women and 45% for men.

Table 6 presents the main results of the hierarchical regression analysis predicting well-being in older women and in Table 7, in older men. No sociodemographic variable was a statistically significant predictor of well-being in either women or men in the post-pandemic period. Furthermore, in the men’s group, perceived vulnerability to infection, which was added in Model 2 did not significantly increase R^2^. Although this variable was statistically significant in the women’s group in Models 2 and 3, it was no longer significant in Model 4 when stress resilience was added to the regression equation. For both genders, the addition of the number of stressful events (Model 3) and stress resilience (Model 4) statistically significantly increased the prediction of well-being. The addition of self-esteem and social support in Model 5 also significantly increased the prediction of well-being, although the increase was greater in the men’s group (R^2^ Change = 0.33) than in the women’s group (R^2^ Change = 0.23). In the final model, with all variables in the regression equation, the main predictor of well-being for women was social support, followed by self-esteem, stress resilience, and the number of stressful events. The main predictor of well-being for men was self-esteem, followed by social support, the number of stressful events, and stress resilience. Older women and men with higher well-being in the post-pandemic period were those with higher self-esteem, perceived social support, greater stress resilience, and those who experienced fewer stressful events. The percentage of total variance explained in well-being was 57% for women and 63% for men.

## 4. Discussion

The overall objective of this study was to investigate the relevance of the COVID-19 pandemic on the stress, mental distress, and well-being of the elderly in Spain. To this end, a repeated cross-sectional study was conducted, analyzing three phases of the pandemic: (1) The first year of the pandemic, between June and December 2020. (2) During the last pandemic year, between February 2022 and February 2023. (3) In the post-pandemic period, between October 2023 and May 2024. Although SARS-CoV-2 has become endemic and many cases are asymptomatic, COVID-19 may be associated with an increase in in-hospital mortality, particularly in older patients [6]. In addition, COVID-19 disease can continue to have a significant clinical impact in older patients with frailty and multimorbidity [4]. Although the older population was considered a particularly vulnerable group during the COVID-19 pandemic [24,25,26,27] and was one of the first groups to experience restrictions on face-to-face contact [25], the entire population was affected by this pandemic, whether infected or not, as the restrictions imposed to prevent the spread of the virus were widespread and affected day life, with repercussions on the mental health and well-being of the population, as well as on the economy and employment [21,22,23]. For this reason, the present study included a comparison group of people aged 30–45 years. Their level of education was controlled to be similar to that of the older people.

The findings of this study suggest that the COVID-19 pandemic increased stress and mental distress in the Spanish population, as has been seen in other countries [38,39,40]. However, some differences were observed according to gender, whether older or adult, and the phase of the pandemic studied. During the first year of the pandemic, half of the older women had experienced mental distress, whereas only a little more than one-third of the older men had experienced distress. In addition, during the first year of the pandemic, older women had more negative feelings than men. These results are in line with other studies that have found that the impact of the COVID-19 pandemic was greater for women than for men [32,35,36,37,41,42,43]. However, it was not only gender that was relevant to mental distress and negative feelings during the first year of the pandemic but also age group, with adult people having more negative feelings than older people. And older people did not have more mental distress than younger people. In fact, rates of mental distress were lower than those of adults of the same gender, both during the pandemic and in the post-pandemic period. These findings are consistent with previous research that found less mental distress in people aged 60 years and older compared with younger people [41]. These findings contribute to the debate on the vulnerability of older adults and suggest that although older people are medically vulnerable, they are resilient and well equipped to cope with a challenging situation [24].

While the most common stressful event for all participants was the illness of family members or other loved ones, reported by nearly half of the participants, the second most common stressful event reported by older people was the death of one or more family members and/or other loved ones, reported by more than one-third of older people but only one-quarter of adults. Older people were also more likely than adults to report their own illness and other stressors, most related to specific physical or mental health problems. In contrast, financial difficulties, serious family discord, serious arguments with their partner, and job loss were more common stressors for adults than for older people. This suggests, as others have noted [53], that the COVID-19 pandemic poses different challenges at different stages of life. This may also explain why, in addition to the differences found between the older and adult groups in the present study, some of the gender differences found differed between the older and adult groups. In particular, during the first year of the pandemic, older women had significantly higher rates of mental distress than men, which was not the case for adults, whereas during the post-pandemic period, the rates of mental distress among adult women (48.5%) were significantly higher than those of adult men (26.5%), whereas the rates of mental distress among older women and older men were very similar (30.2% and 29.8%, respectively). Although the reason for this is unknown, one explanation could be that stress among adult women was significantly higher in the post-pandemic period than in the first year of the pandemic, as there is evidence that stress is linked to negative health outcomes [75]. In addition, serious family discord and serious arguments with their partner were stressors reported more frequently by adult women than by adult men. Older women also reported a greater number of stressful events in the post-pandemic period than older women and men during the first year of the pandemic, although most of these events were related to personal illness, which was more common among older women than among older men. Although the differences were not statistically significant, older men were also less likely to report personal illness in the first year of the pandemic than in the other phases studied. These findings are consistent with evidence that COVID-19 has long-term sequelae and that a significant number of patients who recover from COVID-19 infection continue to have persistent symptoms or develop new symptoms [9,11,12,13].

When analyzing the positive aspects of mental health, such as positive feelings, life satisfaction, thriving, and self-esteem, there were no statistically significant differences between women and men and between older and adult people in the three pandemic phases analyzed. This suggests that the pandemic did not have a major impact on people’s well-being and self-esteem. The impact of gender, age group, and pandemic phase on social support also appeared to be limited. However, adult women reported greater perceived social support than older women during the last year of the pandemic. Although adult men also reported higher levels of social support than older men during this phase, the differences did not reach statistical significance. Adults of both genders also reported greater perceived social support than older people during the post-pandemic period, although the differences were also not statistically significant. These findings are important because there is evidence that social support was a protective factor for mental health during the COVID-19 pandemic [48,49,50,51,76].

The analyses of risk and protective factors for mental distress and well-being for older women and men in the post-pandemic period showed that although both genders had similar risk and protective factors, there were some differences, particularly for mental distress. For both genders, sociodemographic variables (education level, number of children, having a partner, and age) were not significant predictors of mental distress or well-being, except for age, which appeared to be a protective factor for mental distress in men, with older men having less mental distress. One notable gender difference was perceived vulnerability to infection, which was a significant risk factor for distress for women but not for men. In addition, high self-esteem was a protective factor against mental distress for men but not for women. These findings contrast with previous studies during the coronavirus pandemic, where high self-esteem was also found to be a protective factor for mental distress in women [50,51,63,76]. Although the reason for these discrepancies is unknown, it should be noted that these studies did not specifically focus on older women. In addition, the studies cited were conducted during the pandemic period, and these results were obtained in the post-pandemic period. In any case, future work should examine differences in risk and protective factors for mental distress between women and men in the post-pandemic period. Risk and protective factors for well-being were the same for women and men, although their relative importance differed. Whereas the most important protective factor for women was high social support, for men it was high self-esteem. These findings are consistent with traditional socialization patterns and gender stereotypes that associate men with agentic traits and women with communal traits and values [77,78]. For both genders, a greater number of stressful events was a risk factor for well-being, although its relevance was higher for women than for men. Greater resilience to stress was a protective factor for both women and men, although its relevance was higher for men than for women. These findings are consistent with previous studies [46,50,51,63,76], which have found that stressful events are associated with distress and poorer well-being, while resilience and social support are protective factors.

This study has several limitations. First, it is a cross-sectional study. It does not allow us to establish cause-and-effect relationships. It is a convenience sample in which those who have studied at university are overrepresented. Another limitation is due to the data that were collected via an online survey using a Google Form. Therefore, self-reported data may be subject to bias, which reduces the accuracy of the results in comparison to face-to-face interviews. Also, people with higher education levels may be overrepresented due to recruitment through digital platforms, and individuals with severe mental or physical distress may also be underrepresented because they are less likely to participate in voluntary surveys. Instruments and scales used have not been validated specifically for Spanish older adults. Variables such as financial strain, pre-existing mental health conditions, or perceived pandemic threat, which could contribute significantly to mental health outcomes and increase the variance explained, were not assessed. Although it was controlled for equal numbers of women and men in the study and comparison groups, *n* varied between women and men and between the different phases of the pandemic studied. This may affect the power to detect statistically significant differences. Despite these limitations, the results of the present study represent an advance in knowledge of the impact of the COVID-19 pandemic on mental health and well-being and are relevant to the design of policies, programs, and strategies to promote the health and well-being of older people. Promoting resilience, self-esteem and social support for older people are aspects that should be included in such policies and programs. Among the mitigation strategies to be implemented in times of a pandemic, when resources may be particularly scarce, are telemental health services. These could include the use of psychiatric teleconsultation, telemedicine for psychological therapy, videoconferencing, and online social support networks, in addition to regular telephone calls to older people and other vulnerable groups [79].

## 5. Conclusions

The COVID-19 pandemic increased the risk of stress and mental distress in the Spanish population, although some differences were observed according to gender, whether older or adult, and the phase of the pandemic studied. During the first year of the pandemic, older women had significantly more mental distress and negative feelings than older men, although such differences did not occur during the last year of the pandemic and in the post-pandemic period. During the first year of the pandemic, older men had experienced less psychological distress and fewer negative feelings than young adult men, differences that did not occur in the post-pandemic period, whereas older women had experienced less psychological distress and fewer negative feelings than adult women in the post-pandemic period. Our findings contribute to the debate on the vulnerability of older adults and suggest that, although medically vulnerable, they are resilient. Public policies and programs that promote resilience, social support, and self-esteem in older people should be developed, given the importance of such variables for the well-being of older people in the post-pandemic period. As perceived vulnerability to infection is an important predictor of mental distress for older women, programs targeting older women should also aim to reduce their perceived vulnerability to infection.

## Figures and Tables

**Figure 1 geriatrics-10-00005-f001:**
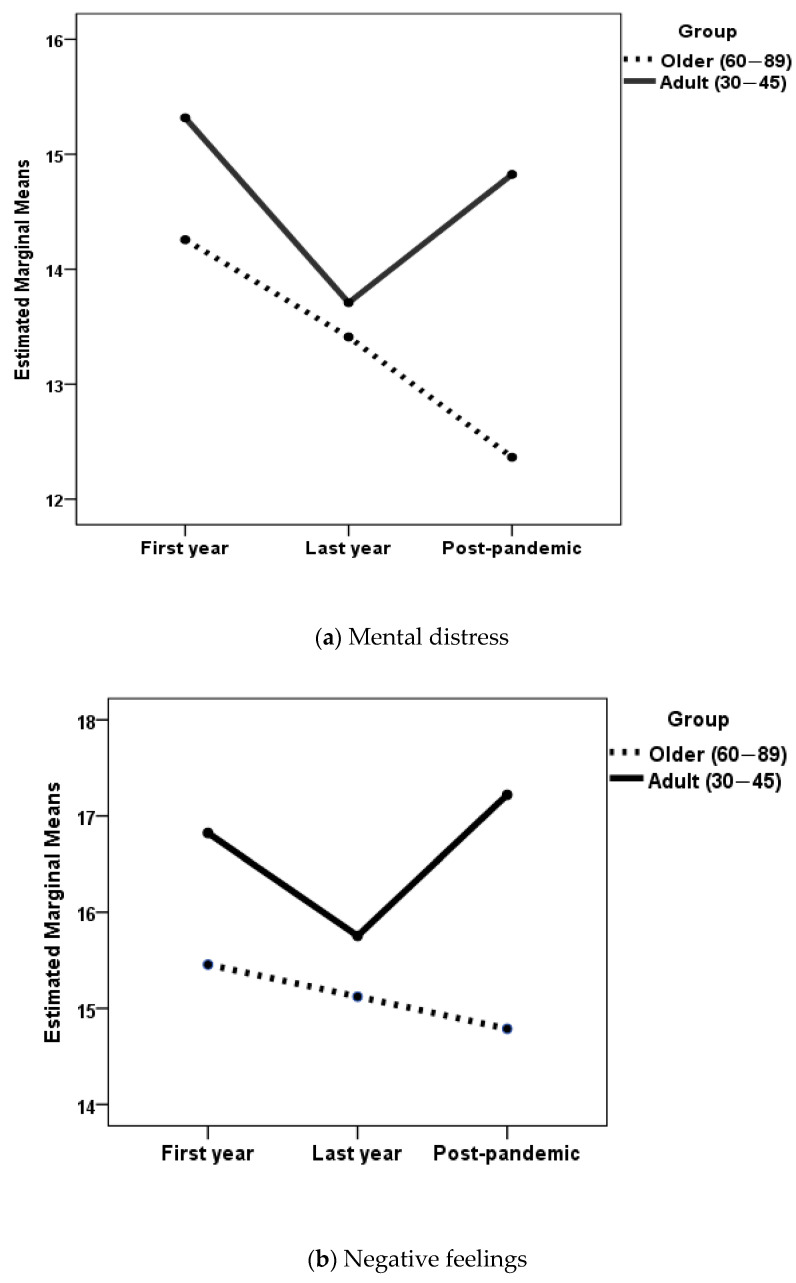
Changes in the study variables as a function of the age group in women.

**Figure 2 geriatrics-10-00005-f002:**
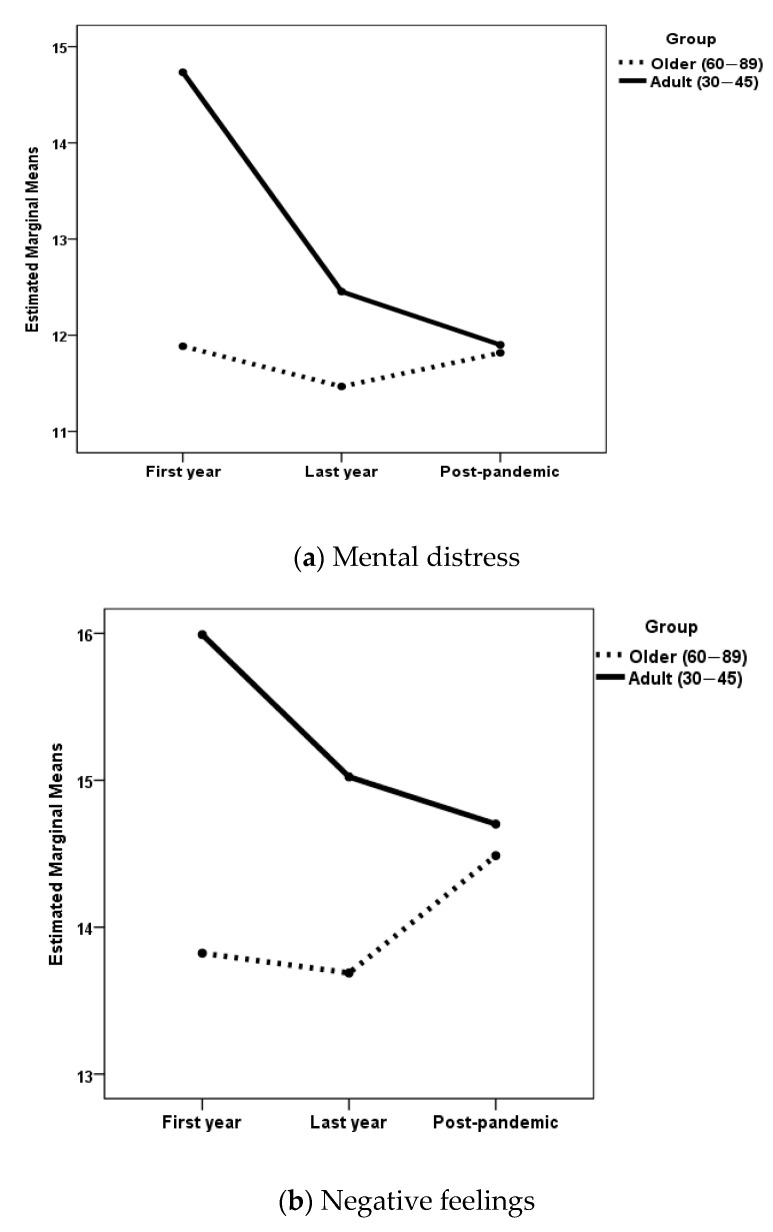
Changes in the study variables as a function of the age group in men.

**Table 1 geriatrics-10-00005-t001:** Sociodemographic characteristics of older groups.

	Men (*n* = 278)		Women (*n* = 440)		χ^2^-Value
	*n*	%	*n*	%	
** *Education* **					
Primary	63	22.9	97	22.1	9.16 *
Secondary	69	25.1	72	16.4	
University	143	52.0	270	61.5	
* **Occupation** *					
Retired	164	59.2	247	56.9	2.16
Unemployed	16	5.8	35	8.1	
Working	90	32.5	136	31.3	
Other	7	2.5	16	3.7	
*Marital status*					
Never married	8	2.9	68	15.7	50.89 ***
Married/partnered	212	76.8	228	52.5	
Separated/divorced	39	14.1	85	19.6	
Widowed	17	6.2	53	12.2	
	*M*	*SD*	*M*	*SD*	*t*-value
Age	66.01	5.77	65.26	5.14	1.78
Number of children	2.28	1.33	1.82	1.30	4.57 ***

Note: * *p* < 0.05; *** *p* < 0.001.

**Table 2 geriatrics-10-00005-t002:** Means (M), standard deviations (SD) and two-way ANOVA statistics for mental distress and well-being in older people (60 to 89 years).

Variable	Men		Women		ANOVA		
	*M*	*SD*	*M*	*SD*	Effect	*F Ratio*	η_p_^2^
*Mental distress*							
First year of the pandemic	11.88	5.82	14.26	6.52			
Last pandemic year	11.47	4.58	13.41	6.11	Gender	11.77 ***	0.016
Post-pandemic period	11.82	5.28	12.36	5.05	Phase	1.82	0.005
Gender × Phase interaction					G × P	1.53	0.004
*Negative feelings*							
First year of the pandemic	13.82	4.43	15.45	4.35			
Last pandemic year	13.69	3.95	15.12	4.20	Gender	11.24 ***	0.016
Post-pandemic period	14.49	3.92	14.79	3.51	Phase	0.19	0.001
Gender × Phase interaction					G × P	1.68	0.005
*Positive feelings*							
First year of the pandemic	20.91	3.74	20.85	4.22			
Last pandemic year	21.38	3.98	21.00	4.29	Gender	0.25	0.000
Post-pandemic period	21.56	4.09	21.50	3.45	Phase	1.63	0.005
Gender × Phase interaction					G × P	0.09	0.000
*Affect balance*							
First year of the pandemic	7.08	7.17	5.39	7.76			
Last pandemic year	7.69	7.10	5.88	7.39	Gender	4.73 *	0.007
Post-pandemic period	7.07	7.43	6.71	6.22	Phase	0.58	0.002
Gender × Phase interaction					G × Phase	0.63	0.002
*Life satisfaction*							
First year of the pandemic	24.74	5.85	24.96	5.64			
Last pandemic year	24.14	5.91	23.92	6.07	Gender	0.26	0.000
Post-pandemic period	25.15	6.17	24.42	5.74	Phase	1.10	0.003
Gender × Phase interaction					G × Phase	0.40	0.001
*Thriving*							
First year of the pandemic	37.85	4.76	37.15	5.74			
Last pandemic year	36.95	5.40	37.58	6.06	Gender	0.04	0.000
Post-pandemic period	37.88	5.83	37.67	5.00	Phase	0.38	0.001
Gender × Phase interaction					G × Phase	0.75	0.002
*Self-esteem*							
First year of the pandemic	21.36	3.78	21.18	4.14			
Last pandemic year	21.72	4.29	21.26	4.87	Gender	0.56	0.001
Post-pandemic period	21.70	4.39	21.56	4.07	Phase	0.45	0.001
Gender × Phase interaction					G × Phase	0.07	0.000
*Social support*							
First year of the pandemic	24.18	8.10	24.70	7.85			
Last pandemic year	23.23	8.39	23.40	7.87	Gender	0.78	0.001
Post-pandemic period	23.22	7.79	24.24	8.02	Phase	1.15	0.003
Gender × Phase interaction					G × Phase	0.13	0.000
*Number of stressful events*							
First year of the pandemic	1.35	1.50	1.27	1.42			
Last pandemic year	1.95	1.27	2.07	1.51	Gender	0.92	0.001
Post-pandemic period	1.82	1.43	2.11	1.46	Phase	17.24 ***	0.049
Gender × Phase interaction					G × Phase	1.00	0.003

Note: * *p* < 0.05; *** *p* < 0.001.

**Table 3 geriatrics-10-00005-t003:** Means (M), standard deviations (SD) and two-way ANOVA statistics for mental distress and well-being in adult people (30 to 45 years).

Variable	Men		Women		ANOVA		
	*M*	*SD*	*M*	*SD*	Effect	*F Ratio*	η_p_^2^
*Mental distress*							
First year of the pandemic	14.73	7.34	15.32	6.57			
Last pandemic year	12.45	6.11	13.71	6.82	Gender	8.69 **	0.012
Post-pandemic period	11.90	5.84	14.82	7.03	Phase	5.98 **	0.017
Gender × Phase interaction					G × P	1.91	0.005
*Negative feelings*							
First year of the pandemic	15.99	4.58	16.82	4.38			
Last pandemic year	15.02	4.37	15.75	4.55	Gender	14.65 ***	0.020
Post-pandemic period	14.70	4.15	17.22	4.32	Phase	2.78	0.008
Gender × Phase interaction					G × P	2.71	0.008
*Positive feelings*							
First year of the pandemic	20.84	4.54	20.82	4.02			
Last pandemic year	21.98	5.05	22.47	4.14	Gender	0.15	0.000
Post-pandemic period	22.37	4.06	21.50	4.86	Phase	6.80 **	0.019
Gender × Phase interaction					G × P	1.15	0.003
*Affect balance*							
First year of the pandemic	4.85	8.19	3.99	7.58			
Last pandemic year	6.95	8.44	6.71	7.99	Gender	5.57 *	0.008
Post-pandemic period	7.67	7.17	4.28	8.18	Phase	5.47 **	0.015
Gender × Phase interaction					G × Phase	2.23	0.006
*Life satisfaction*							
First year of the pandemic	24.26	7.05	23.90	6.55			
Last pandemic year	25.46	6.67	24.35	6.61	Gender	4.31 *	0.006
Post-pandemic period	25.39	6.41	23.45	7.39	Phase	0.77	0.002
Gender × Phase interaction					G × Phase	0.85	0.002
*Thriving*							
First year of the pandemic	37.34	7.26	37.47	6.40			
Last pandemic year	39.26	6.83	39.01	6.75	Gender	0.38	0.001
Post-pandemic period	38.73	7.09	37.84	7.02	Phase	3.42 *	0.010
Gender × Phase interaction					G × Phase	0.34	0.001
*Self-esteem*							
First year of the pandemic	20.92	4.76	20.98	4.95			
Last pandemic year	21.93	5.51	21.52	5.26	Gender	0.57	0.001
Post-pandemic period	21.25	4.93	20.65	5.94	Phase	1.32	0.004
Gender × Phase interaction					G × Phase	0.27	0.001
*Social support*							
First year of the pandemic	24.25	9.04	26.21	8.46			
Last pandemic year	26.12	8.13	27.47	8.72	Gender	3.72	0.005
Post-pandemic period	26.15	8.57	26.81	7.68	Phase	2.27	0.006
Gender × Phase interaction					G × Phase	0.36	0.001
*Number of stressful events*							
First year of the pandemic	1.83	1.59	1.59	1.45			
Last pandemic year	1.95	1.51	1.97	1.66	Gender	0.28	0.000
Post-pandemic period	1.81	1.45	2.22	1.56	Phase	2.72	0.008
Gender × Phase interaction					G × Phase	2.65	0.008

Note: * *p* < 0.05; ** *p* < 0.01; *** *p* < 0.001.

**Table 4 geriatrics-10-00005-t004:** Summary of the hierarchical regression results for mental distress for older women.

	Model 1		Model 2		Model 3		Model 4		Model 5	
Variable	β	*t*-Value	β	*t*-Value	β	*t*-Value	β	*t*-Value	β	*t*-Value
Age	0.06	0.52	0.08	0.80	0.05	0.54	0.03	0.32	−0.01	−0.07
Number of children	−0.05	−0.40	−0.10	−0.91	−0.10	−0.99	−0.07	−0.69	−0.05	−0.56
Education	−0.11	−0.93	−0.05	−0.49	−0.07	−0.78	−0.08	−0.85	−0.11	−1.22
Married/partnered	−0.05	−0.49	−0.09	−0.92	0.00	0.01	0.02	0.27	0.03	0.39
Perceived vulnerability to infection			0.50	5.27 ***	0.36	3.91 ***	0.32	3.64 ***	0.29	3.31 **
Number of stressful events					0.38	4.11 ***	0.34	3.90 ***	0.30	3.36 **
Stress resilience							−0.30	3.62 ***	−0.24	−2.85 **
Self-esteem									−0.12	−1.28
Social support									−0.16	−1.72
R^2^	0.02		0.26		0.38		0.46		0.51	
Adjusted R^2^	−0.03		0.21		0.34		0.42		0.46	
R^2^ Change	0.02		0.24 ***		0.12 ***		0.08 ***		0.05 *	

Note: β = standardized regression coefficient. R^2^ = explained variance. * *p* < 0.05; ** *p* < 0.01; ****p* < 0.001.

**Table 5 geriatrics-10-00005-t005:** Summary of the hierarchical regression results for mental distress for older men.

	Model 1		Model 2		Model 3		Model 4		Model 5	
Variable	β	*t*-Value	β	*t*-Value	β	*t*-Value	β	*t*-Value	β	*t*-Value
Age	−0.26	−2.45 *	−0.25	−2.44*	−0.26	−2.86 **	−0.27	−3.15 **	−0.21	−2.68 **
Number of children	0.14	1.21	0.13	1.15	0.12	1.19	0.14	1.49	0.08	0.91
Education	0.01	0.10	−0.00	−0.01	0.03	0.31	0.05	0.59	0.03	0.31
Married/partnered	0.01	0.11	0.02	0.19	−0.01	−0.09	0.03	0.31	0.00	0.01
Perceived vulnerability to infection			0.21	2.19*	0.16	1.82	0.03	0.28	0.02	0.26
Number of stressful events					0.42	4.78 ***	0.35	4.14 ***	0.33	4.15 ***
Stress resilience							−0.36	−3.88 ***	−0.20	−2.14 *
Self-esteem									−0.33	−3.63 ***
Social support									−0.10	−1.18
R^2^	0.06		0.11		0.28		0.38		0.50	
Adjusted R^2^	0.02		0.06		0.24		0.34		0.45	
R^2^ Change	0.06		0.05*		0.18 ***		0.10 ***		0.11 ***	

Note: β = standardized regression coefficient. R^2^ = explained variance. * *p* < 0.05<; ** *p* < 0.01; *** *p* < 0.001.

**Table 6 geriatrics-10-00005-t006:** Summary of the hierarchical regression results for well-being for older women.

	Model 1		Model 2		Model 3		Model 4		Model 5	
Variable	β	*t*-Value	β	*t*-Value	β	*t*-Value	β	*t*-Value	β	*t*-Value
Age	−0.01	−0.03	−0.02	−0.16	0.01	0.08	0.04	0.41	0.12	1.50
Number of children	0.01	0.04	0.04	0.34	0.04	0.36	−0.01	−0.08	−0.04	−0.42
Education	0.01	0.08	−0.03	−0.28	−0.01	−0.11	−0.01	−0.11	0.06	0.77
Married/partnered	0.08	0.70	0.10	0.98	0.03	0.26	−0.00	−0.05	−0.02	−0.32
Perceived vulnerability to infection			−0.35	−3.43 **	−0.24	−2.27 *	−0.18	−1.86	−0.10	−1.35
Number of stressful events					−0.32	−3.08 **	−0.27	−2.81 **	−0.16	−2.01 *
Stress resilience							0.42	4.67 ***	0.28	3.73 ***
Self-esteem									0.29	3.40 **
Social support									0.34	4.10 ***
R^2^	0.01		0.13		0.21		0.38		0.61	
Adjusted R^2^	−0.04		0.08		0.16		0.32		0.57	
R^2^ Change	0.01		0.12 **		0.09 **		0.16 ***		0.23 ***	

Note: β = standardized regression coefficient. R^2^ = explained variance. * *p* < 0.05, ** *p* < 0.01; *** *p* < 0.001.

**Table 7 geriatrics-10-00005-t007:** Summary of the hierarchical regression results for well-being for older men.

	Model 1		Model 2		Model 3		Model 4		Model 5	
Variable	β	*t*-Value	β	*t*-Value	β	*t*-Value	β	*t*-Value	β	*t*-Value
Age	0.07	0.69	0.07	0.66	0.08	0.83	0.09	1.03	0.00	0.01
Number of children	−0.08	−0.73	−0.08	−0.69	−0.07	−0.67	−0.10	−1.01	0.02	0.24
Education	−0.11	−0.99	−0.11	−0.95	−0.14	−1.28	−0.17	−1.74	−0.10	−1.46
Married/partnered	−0.02	−0.22	−0.03	−0.25	−0.00	−0.03	−0.05	−0.53	−0.01	−0.18
Perceived vulnerability to infection			−0.10	−0.96	−0.05	−0.54	0.12	1.32	0.13	1.95
Number of stressful events					−0.38	−3.91 ***	−0.28	−3.18 **	−0.24	−3.81 ***
Stress resilience							0.46	4.81 ***	0.20	2.67 **
Self-esteem									0.51	6.80 ***
Social support									0.25	3.68 ***
R^2^	0.02		0.03		0.16		0.33		0.66	
Adjusted R^2^	−0.02		−0.02		0.11		0.28		0.63	
R^2^ Change	0.02		0.01		0.14 ***		0.17 ***		0.33 ***	

Note: β = standardized regression coefficient. R^2^ = explained variance. ** *p* < 0.01; *** *p* < 0.001.

## Data Availability

Data are available from the author upon reasonable request.

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
