# Peer review of "COVID-19 and Mental Distress and Well-Being Among Older People: A Gender Analysis in the First and Last Year of the Pandemic and in the Post-Pandemic Period"

_geriatrics, 2025, doi:10.3390/geriatrics10010005_

Round 1
Reviewer 1 Report
Comments and Suggestions for Authors
The manuscript "COVID-19 and mental distress and well-being among older people: A gender analysis in the first and last year of the pandemic and in the post-pandemic period" is under review. This manuscript investigates the relevance of the COVID-19 pandemic on stress, mental distress and well-being of older people in Spain.
The manuscript is of high quality and relevance. The large sample used (1436 participants), the creation of two well differentiated groups (30 to 45 and 60 to 89 years) and the existence of three evaluation phases (first year of the pandemic, last year of the pandemic and post-pandemic period) are particularly appreciated. In addition, the variables studied are numerous and relevant. I consider the data analysis to be adequate and complete, and the figures included are very clear and very helpful in understanding the text. The discussion is also adequate and the references are well chosen. Therefore, I consider this to be a good manuscript.
Since all papers can be improved, I would like to specify a number of points that may require minor changes that, in my opinion, may help to improve the manuscript:
11) What is the post-pandemic period? From 5 May 2023, when the pandemic period is officially declared over, which period should be chosen? It was decided to choose dates between October 2023 and May 2024. However, the post-covid period ends in February 2023, so these periods almost overlap. I think it would be good to clarify in the text why the cut-off dates for each period were chosen, and to recognise the possible overlap of dates as a limitation of the study.
22) It is stated that the coronavirus emerges in Wuhan (China). I think this is very controversial and no matter how much research has been done, it has not been established with certainty. I recommend that this statement be removed.
33) In Table 1, I think the "missing data" line is not necessary and can be explained in the previous or subsequent text.
44) I think it is more appropriate to include an example item for each instrument.
Author Response
Dear Reviewer 1,
Thank you for your valuable feedback and insightful comments. Please find below our detailed response to each recommendation. In the revised text, all changes are highlighted in red.
The manuscript "COVID-19 and mental distress and well-being among older people: A gender analysis in the first and last year of the pandemic and in the post-pandemic period" is under review. This manuscript investigates the relevance of the COVID-19 pandemic on stress, mental distress and well-being of older people in Spain.
The manuscript is of high quality and relevance. The large sample used (1436 participants), the creation of two well differentiated groups (30 to 45 and 60 to 89 years) and the existence of three evaluation phases (first year of the pandemic, last year of the pandemic and post-pandemic period) are particularly appreciated. In addition, the variables studied are numerous and relevant. I consider the data analysis to be adequate and complete, and the figures included are very clear and very helpful in understanding the text. The discussion is also adequate and the references are well chosen. Therefore, I consider this to be a good manuscript.
Since all papers can be improved, I would like to specify a number of points that may require minor changes that, in my opinion, may help to improve the manuscript:
1) What is the post-pandemic period? From 5 May 2023, when the pandemic period is officially declared over, which period should be chosen? It was decided to choose dates between October 2023 and May 2024. However, the post-covid period ends in February 2023, so these periods almost overlap. I think it would be good to clarify in the text why the cut-off dates for each period were chosen, and to recognise the possible overlap of dates as a limitation of the study.
Response: Thank you for your comment. It has allowed us to clarify the dates of data collection and why these dates were chosen for the study. In the revised manuscript, on page 3, lines 117 to 142, we have expanded the description of each of the phases in which data were collected and the reasons for selecting these phases. As explained in the revised manuscript, since there is no generally accepted quantitative definition of the end of a pandemic, as in the case of COVID-19, in this study the post-pandemic period is considered to be the period following the declaration of the end of COVID-19 as an international public health emergency by the WHO on 5 May 2023 and the declaration of the end of the health crisis situation caused by COVID-19 in Spain by the Council of Ministers at its meeting on 4 July 2023. In addition, three months were waited from the latter date to start the data collection to give the population time to become aware of the new situation. Specifically, this data collection took place between October 2023 and May 2024. The other two data collections took place during the first pandemic year of COVID-19, between June and December 2020, and during the last pandemic year, between February 2022 and February 2023. Therefore, there is no overlap of dates.
2) It is stated that the coronavirus emerges in Wuhan (China). I think this is very controversial and no matter how much research has been done, it has not been established with certainty. I recommend that this statement be removed.
Response: In the revised manuscript, it was removed that the coronavirus emerged in Wuhan, China.
3) In Table 1, I think the "missing data" line is not necessary and can be explained in the previous or subsequent text.
Response: In the revised manuscript, the "missing data" line has been removed from Table 1.
4) I think it is more appropriate to include an example item for each instrument.
Response: In the revised manuscript, an example item has been included for each instrument.
We are grateful to Reviewer 1 for all his/her suggestions and comments, as they have contributed to the improvement of the quality of the study.
Reviewer 2 Report
Comments and Suggestions for Authors
This manuscript is a potential interest in understanding the connection between isolation and wellbeing in older adults.
Recommendations are provided to the authors in the attached file to improve the clarity of the presentation and to be provide more specific policy recommendations.

Author Response
Dear Reviewer 2,
Thank you for your valuable feedback and insightful comments. Please find below our detailed response to each recommendation. In the revised text, all changes are highlighted in red.
In addition to the responses in this document, please find attached the pdf with your revision. In the pdf document you kindly sent us with your revisions, we have responded to all of your suggestions and comments, but we have not been able to include the changes that were made because we had to make the changes in the edited document that the editor of the journal sent us.
- world's population
Response: It does not seem appropriate to include “world’s population” in the abstract, as suggested by Reviewer 2, as the study was conducted in Spain.
- Use the age friendly term: "older adults" in the abstract and throughout the manuscript, including tables.
Response: Although I consider your suggestion to be very valuable, the term "older adults" has not been used in the abstract or throughout the manuscript because, since the other research group is of adults, we feel that inclusion of the term "adults" in both groups would be confusing.
- Describe the population sample
Response: The description of the sample has been slightly expanded on the abstract, but it is not possible to expand it further as the length of the abstract would be much longer than the journal allows.
- Older adults
Response: As mentioned above, to avoid using the term “adults” in both groups, the term “older adults” has not been used.
- Please describe differences between the older and younger populations as indicated in figures 1 and 2
Response: As the abstract is longer than the journal rules allow, it has not been possible to include the results of Figures 1 and 2. In any case, the most important results of these figures, i.e. those related to psychological distress, are already included in the abstract when describing the rates of psychological distress (in lines 15 to 19).
- Make specific policy recommendations
Response: Policy recommendations could not be made in the abstract because, as noted above, it is longer than the journal allows. However, specific policy recommendations were made on page 21, at the end of the Discussion section.
- Lines 81-90 are unclear
Response: The text of lines 81-90 of the original manuscript, which correspond to lines 97-107 in the revised manuscript, has been changed for clarity.
- Phrase this differently: affected men and with min differently
Response: We have changed the phrase as suggested by Reviewer 2 in the revised manuscript (on line 114).
- Lines 102-114 are nuclear
Response: The text corresponding to lines 102 to 114 of the first version of the manuscript has been changed in the revised manuscript (in lines 144 to 153) to make it clearer.
- Describe subject recruitment early in the design.
Response: In the revised manuscript, subject recruitment has been placed early in the design, on page 4, lines 158 to 162.
- Lines 128-130 unclear
Response: The text corresponding to lines 128 to 130 of the first version of the manuscript has been changed in the revised manuscript (in lines 170 to 175) for clarity.
- Older adults
Response: As mentioned above, to avoid using the term “adults” in both research groups, the term “older adults” has not been used.
- Please format to clarify the subheadings. Also define the term "affect balance".
Response: In the revised manuscript, the subheadings have been bolded for clarity. The term “affect balance” has also been defined (on page 5 lines 225 to 227).
- Insert a table of the tools utilized
Response: Although we made some changes to the instruments used, we did not include a table of the tools utilized because the journal instructions do not specify that the tools used must be in a table.
- This section should be shorter, cite only p<.05, site only significant comparisons in the narrative and do not and certain statistics which are present in the table.
Response: In the revised manuscript, the section referred to by the Reviewer 2 has been shortened by including chi-square values only when p<.05, and the statistics given in the text do not are present in the tables.
- Placed tables 4 through 7 and an appendix
Response: Tables 4 to 7 have been kept in the text rather than in an appendix as they are fundamental to the work carried out. They correspond to the analyses carried out to meet the second aim of the study, and the text of section 3.2. is much better understood with the tables close to the text.
- Explained the differences found between older and younger subjects as displayed in figures 1 and 2.
Response: The results of Figures 1 and 2 are explained in the results section on pages 11, 13, 14 and 15. In the discussion, we have only commented on the results that we consider most relevant, and we have done so in an integrated way with the rest of the results of the study.
- Have another paragraph as the conclusion and give specific policy recommendations based on the findings
Response: In the revised manuscript, the conclusions have been placed in a separate paragraph and specific recommendations have been added at the end of the Discussion section, on page 21.
We are grateful to Reviewer 2 for all her suggestions and comments, as they have contributed to the improvement of the quality of the study.

Reviewer 3 Report
Comments and Suggestions for Authors
1. The timing of data collection across the three pandemic phases is vaguely described. The "first pandemic year" and "last pandemic year" overlap globally recognized pandemic waves. The authors should provide further explanation and justification for these time periods.
2. Did the Google survey form guard against duplicate responses, e.g., IP filtering? There is also no mention of how missing data was handled, particularly given the likelihood of incomplete responses in online surveys.
3. For the section on study variables and instruments, these scales, although validated in other settings, may not be culturally or contextually validated for Spanish older adults. This should be acknowledged.
4. There is insufficient detail on the baseline equivalence of groups.
5. The sample sizes for the three pandemic phases are uneven (702, 326, and 408, respectively), which may affect the reliability of statistical comparisons. This imbalance is not adequately justified or addressed through weighted analyses.
6. For the analysis plan, it would be better for the author to implement a difference-in-difference (DiD) analysis. Differences between the older and younger groups (e.g., sociodemographics) might bias results. The study also analyzes mental distress, well-being, and other variables across three pandemic phases (first year, last year, and post-pandemic period) without controlling for external temporal effects that could influence outcomes. A DiD approach would better account for these time-related confounders by comparing changes over time between treated (older adults) and control (younger adults) groups.
7. The β values, although significant, explain only a limited proportion of the variance. Variables such as financial strain, access to social support, pre-existing mental health conditions, or perceived pandemic threat might be significant contributors to mental health outcomes but are not included in the model.
8. In the discussion section, as part of mitigation strategies, as resources could be particularly scarce during a serious pandemic situation, timely psychological support could also take many forms, including telemedicine and informal support groups (citation: pubmed.ncbi.nlm.nih.gov/32380875) especially for older and vulnerable adults. This should be mentioned.
9. The discussion of study limitations is inadequate. For example, people with higher education levels may be overrepresented due to recruitment through digital platforms. Individuals with severe mental or physical distress may also be underrepresented because they are less likely to participate in voluntary surveys.
10. Suggest to have a separate conclusions section.
Author Response
Dear Reviewer 3,
Thank you for your valuable feedback and insightful comments. Please find below our detailed response to each recommendation. In the revised text, all changes are highlighted in red.
- The timing of data collection across the three pandemic phases is vaguely described. The "first pandemic year" and "last pandemic year" overlap globally recognized pandemic waves. The authors should provide further explanation and justification for these time periods.
Response: In the revised manuscript, on page 3, lines 117-142, we have expanded the description of each of the phases in which data were collected and the justification for these time periods. In the present study, it was decided not to use pandemic waves because, in addition to the fact that the chronology of such waves varied between countries, their relevance and frequency were much greater in the first two years of the pandemic. Specifically, in Spain, there were 6 waves from the beginning of the COVID-19 pandemic until March 28, 2022, when a new surveillance strategy came into effect. Much research has been done on pandemic waves in different countries around the world. However, this was not the focus of this study.
- Did the Google survey form guard against duplicate responses, e.g., IP filtering? There is also no mention of how missing data was handled, particularly given the likelihood of incomplete responses in online surveys.
Response: In order to protect the anonymity of the respondents as much as possible, and thus increase their participation and the sincerity of their responses, the Google survey did not ask for any data that would allow them to be identified. However, the responses were extensively analysed in databases, first in the Excel database where the Google form collected the responses, and then in the SPSS database where they were analysed. This analysis focused both on looking for possible duplicate responses and on looking for responses that indicated that the survey had not been conducted in a rigorous manner. Although the only mandatory response in the survey was the informed consent, it was explained at the beginning of the survey that it was very important to answer all the questions. In addition, all those who participated in the sample collection by sending the link to their social networks were told that the data was for a scientific study and that it was important that the link was sent to people who knew that they were being surveyed seriously. In addition, in the case of students receiving course credit, they were warned that they would only receive credit if the interviews were completed in full and no evidence of frivolous responses was found.
The number of missing data was very low, never exceeding 2%. Such responses were replaced by the mean in the ANOVA and multiple regression analyses, but in the descriptive and chi-square analyses only those individuals who were not missing for the variable analysed were included.
- For the section on study variables and instruments, these scales, although validated in other settings, may not be culturally or contextually validated for Spanish older adults. This should be acknowledged.
Response: In the revised manuscript, on page 21 lines 719-720, the limitations acknowledge that the instruments and scales used have not been validated specifically for Spanish older adults.
- There is insufficient detail on the baseline equivalence of groups.
Response: Given that the study does not propose to implement or analyse the effect of an intervention, and that there is no experimental and control group, we believe that there is no need for equivalence of the groups, except for the level of education, which has been controlled for. Indeed, given that it is necessary for the groups to belong to different age groups, equivalence in many of the socio-demographic characteristics is not expected. For example, older people are expected to be more likely to be retired and widowed than people aged 30-45. And knowing whether there are differences in stress, mental distress and well-being between older and adult groups in the three phases of the COVID-19 pandemic studied is part of the first aim of the study.
- The sample sizes for the three pandemic phases are uneven (702, 326, and 408, respectively), which may affect the reliability of statistical comparisons. This imbalance is not adequately justified or addressed through weighted analyses.
Response: We agree with your statement that the sample sizes for the three pandemic phases are uneven, which may affect the reliability of statistical comparisons. This was therefore acknowledged as a limitation of the study in the first version of the submitted manuscript and is still included in the limitations of the study in the revised manuscript (on page 21, lines 722 to 725). Working in a pandemic such as COVID-19 is very complicated because there was uncertainty about what was going to happen. In addition, the response rate was higher in women than in men, a phenomenon that occurs in many studies when it comes to voluntary participation. Nevertheless, we preferred to keep the data as they were, rather than weighting or deleting cases to keep the same n in the older group, because there are data whose value is important in itself (e.g. percentages, R2, means or standard deviations).
- For the analysis plan, it would be better for the author to implement a difference-in-difference (DiD) analysis. Differences between the older and younger groups (e.g., sociodemographics) might bias results. The study also analyzes mental distress, well-being, and other variables across three pandemic phases (first year, last year, and post-pandemic period) without controlling for external temporal effects that could influence outcomes. A DiD approach would better account for these time-related confounders by comparing changes over time between treated (older adults) and control (younger adults) groups.
Response: Although your suggestion seems very interesting and I will take it into account in future intervention work, it was not followed in the present study. The design of the present study is not longitudinal and is not intended to test the effect of an intervention. What we want to study is mental distress and well-being among older people in three very different periods of the COVID-19 pandemic (the first year, the last year and the post-pandemic period). Therefore, there is no need to control for external temporal effects, as there is no intervention to test. A comparison group of adults was included in the present work with the purpose of deepening the knowledge of the relevance of the three phases of the pandemic studied on the mental distress and well-being of older people, analysing whether the results found are a phenomenon that occurs in other age groups or whether it is specific to older people.
- The β values, although significant, explain only a limited proportion of the variance. Variables such as financial strain, access to social support, pre-existing mental health conditions, or perceived pandemic threat might be significant contributors to mental health outcomes but are not included in the model.
Response. In the revised manuscript, the limitations section (on page 21 lines 720 to 722) notes that variables such as financial strain, pre-existing mental health conditions, or perceived pandemic threat, which could contribute significantly to mental health outcomes and increase the variance explained, were not assessed. Access to social support was not mentioned in the limitations as one of the study variables was perceived social support.
- In the discussion section, as part of mitigation strategies, as resources could be particularly scarce during a serious pandemic situation, timely psychological support could also take many forms, including telemedicine and informal support groups (citation: pubmed.ncbi.nlm.nih.gov/32380875) especially for older and vulnerable adults. This should be mentioned.
Response: Thank you for your suggestion, the text of the link you provided has been read. In the revised manuscript, your suggestion has been included on page 21 at the end of the Discussion section, along with the reference.
- The discussion of study limitations is inadequate. For example, people with higher education levels may be overrepresented due to recruitment through digital platforms. Individuals with severe mental or physical distress may also be underrepresented because they are less likely to participate in voluntary surveys.
Response: In the revised manuscript, the limitations have been expanded. On page 21, lines 716 to 719, limitations include that people with higher education levels may be overrepresented due to recruitment through digital platforms, and that individual with severe mental or physical distress may also be underrepresented because they are less likely to participate in voluntary surveys.
- Suggest to have a separate conclusions section.
Response: A separate Conclusion section has been included in the revised manuscript.
We are grateful to Reviewer 3 for all her suggestions and comments, as they have contributed to the improvement of the quality of the study.
Reviewer 4 Report
Comments and Suggestions for Authors
This is a well-written and well-organized research article with adequate novelty. Some points should be addressed.
- At the first paragraph of the Introduction section. The last sentence should be enhriched with more details and relevant references.
- In the introduction section, the authors should add a paragraph with the long-term effects of Covid-19 pandemic, including relevant references.
- The last paragraph of the Introduction section reporting the aims of the study is too long. The authos should condense the text of the last paragraph, focusing on the mai purposes of the studying in combination with the literature gap that exists and that the present study will cover.
- At the beggining of the section 2.1, the authors should clearly report if the older adults of the study are the same in the 3 phases of study or if they are different in each phase.
- Are there any references that could be added in the paragraph of lines 172-178? There are several validated questionnaires that the authors could use to evaluate the stress of participants.
- Did the authors use a normality test for continuous variables? This should be stated in Data Analysis section (2.3 section).
- X square value could be deleted from the text of the Results section.
- The resolution of Figures 1 and 2 should be improved in order to be more readable.
- In the limitations of the study the authors should report that their data were collected through an online survey using a Google form. So, self-reported data should be reported to include bias, reducing the accurancy of the resluts in comparison to face-to-face interviews.
- A separate section with the main conclusions of the study should be added.
Author Response
Dear Reviewer 4,
Thank you for your valuable feedback and insightful comments. Please find below our detailed response to each recommendation. In the revised text, all changes are highlighted in red.
This is a well-written and well-organized research article with adequate novelty. Some points should be addressed.
- At the first paragraph of the Introduction section. The last sentence should be enhriched with more details and relevant references.
Response: In the revised manuscript, the last sentence of the first paragraph has been enriched with more details and relevant references (pages 1-2, lines 40 to 50).
- In the introduction section, the authors should add a paragraph with the long-term effects of Covid-19 pandemic, including relevant references.
Response: In the revised manuscript, on page 2, lines 51 to 61, a paragraph on the long-term effects of the Covid-19 pandemic has been added, including relevant references (references number 9 to 14).
- The last paragraph of the Introduction section reporting the aims of the study is too long. The authos should condense the text of the last paragraph, focusing on the mai purposes of the studying in combination with the literature gap that exists and that the present study will cover.
Response: Following your suggestions, the last paragraph of the Introduction section reporting the aims of the study has been corrected in the revised manuscript.
- At the beggining of the section 2.1, the authors should clearly report if the older adults of the study are the same in the 3 phases of study or if they are different in each phase.
Response: In the revised manuscript on section 2.1, page 4, lines 170-171, it is clearly stated that the individuals who responded at each phase are different, as this is a cross-sectional design.
- Are there any references that could be added in the paragraph of lines 172-178? There are several validated questionnaires that the authors could use to evaluate the stress of participants.
Response: In the revised manuscript, three references have been added to the paragraph that was on lines 172-178 in the original manuscript, corresponding to three published articles in which stress during COVID-19 was assessed using the same procedure as in the present study. (See page 5, line 213).
- Did the authors use a normality test for continuous variables? This should be stated in Data Analysis section (2.3 section).
Response: In the revised manuscript, in the Data Analysis section (on page 6, lines 271 to 273), it was included that the normality for continuous variables was analyzed by graphical and numerical methods analyzing kurtosis and skewness, following the suggestions of, among others, Mishra et al. 2019. Homogeneity of variance was checked using Levene's test.
- X square value could be deleted from the text of the Results section.
Response: In the revised manuscript, X square values were deleted from the text of the Results section when the differences in percentages were not statistically significant, although they were retained when the differences were statistically significant.
- The resolution of Figures 1 and 2 should be improved in order to be more readable.
Response: Figures 1 and 2 have been improved in order to be more readable in the revised manuscript.
- In the limitations of the study the authors should report that their data were collected through an online survey using a Google form. So, self-reported data should be reported to include bias, reducing the accurancy of the resluts in comparison to face-to-face interviews.
Response: In the revised manuscript, on page 21, lines 713 to 716, the limitations reports that the data were collected via an online survey using a Google form. Therefore, self-report data may be subject to bias, which reduces the accuracy of the results in comparison to face-to-face interviews.
- A separate section with the main conclusions of the study should be added.
Response: The revised manuscript (on page 21) includes a separate section with the main conclusions of the study.
We are grateful to Reviewer 4 for all her suggestions and comments, as they have contributed to the improvement of the quality of the study.
Round 2
Reviewer 2 Report
Comments and Suggestions for Authors
The authors have made incredible response to reviewer comments. Some recommendations to improve the clarity of the presentation are provided.
Line 24 of the abstract - add: predictors of mental distress for men
Lines 48-50: Delete the quotes and p11
Line 61 - Substitute: long-term management
Line 190 of the Methods: Describe the data base
Lines 194-199 - delete as also described on p. 21
Methods section 2.2: Can you provide these short instruments in an appendix?
Figures 1,2 ; label from the top
Conclusion:
Line 745 -younger adult men
Line 749 -use: older adults
Last sentence -this is repetitive. Can you provide any specific recommendations?
Author Response
We thank Reviewer 2 for all his/her reviews and recommendations, which contributed significantly to the improvement of the manuscript.
The authors have made incredible response to reviewer comments. Some recommendations to improve the clarity of the presentation are provided.
Line 24 of the abstract - add: predictors of mental distress for men
Response: In the corrected manuscript, lines 24 and 25 of the abstract read: Low self-esteem and younger age were also predictors of mental distress for men
Lines 48-50: Delete the quotes and p11
Response: In the corrected manuscript, the quotes and p11 in lines 48-50 have been deleted. However, since the page number and the quotes are there because text was a literal quotation, the text has been paraphrased. In the corrected manuscript, lines 48-50 read: suggest an association between severe COVID-19 and cognitive impairment beyond the acute phase of illness [8].
Line 61 - Substitute: long-term management
Response: In line 61 of the corrected manuscript, long-term management has been placed.
Line 190 of the Methods: Describe the data base
Response: The database was described in lines 190 and 191 of the Methods of the corrected manuscript.
Lines 194-199 - delete as also described on p. 21
Response: Lines 194-199 have been deleted.
Methods section 2.2: Can you provide these short instruments in an appendix?
Response: For copyright reasons, the instruments used cannot be provided in an appendix. However, since all instruments are referenced, interested readers can obtain them.
Figures 1,2 ; label from the top
Response: The labels in Figures 1 and 2 are positioned according to the journal template. Therefore they cannot be placed on the top.
Conclusion:
Line 745 -younger adult men
Response: Response: younger adult men has been placed as suggested by the Reviewer. Since lines 194-199 were deleted in the corrected manuscript, younger adult men is in the Conclusion on line 740.
Line 749 -use: older adults
Response: Older adults has been placed as suggested by the Reviewer. Since lines 194-199 were deleted in the corrected manuscript, older adults is in the Conclusion on line 744.
Last sentence -this is repetitive. Can you provide any specific recommendations?
Response: The last sentence of the conclusion has been deleted. At the end of the conclusion, lines 744-749, recommendations have been added that are intended to be as specific as the results of the study conducted allow.
Reviewer 3 Report
Comments and Suggestions for Authors
Thank you for the replies and revisions.
Author Response
Thank you for the replies and revisions.
Response: Thank you very much for your review and feedback on the revised manuscript.
Reviewer 4 Report
Comments and Suggestions for Authors
The authors have significantly improved their manuscript.
Author Response
The authors have significantly improved their manuscript.
Response: Thank you very much for your review and feedback on the revised manuscript.